# Morning Headache as an Obstructive Sleep Apnea-Related Symptom among Sleep Clinic Patients—A Cross-Section Analysis

**DOI:** 10.3390/brainsci10010057

**Published:** 2020-01-19

**Authors:** Jakub Spałka, Konrad Kędzia, Wojciech Kuczyński, Aleksandra Kudrycka, Aleksandra Małolepsza, Piotr Białasiewicz, Łukasz Mokros

**Affiliations:** 1Department of Sleep Medicine and Metabolic Disorders, Medical University of Lodz, 90-001 Lodz, Poland; jakub.spalka@gmail.com (J.S.); aleksandra.kudrycka@stud.umed.lodz.pl (A.K.); aleksandra.malolepsza1@stud.umed.lodz.pl (A.M.); piotr.bialasiewicz@umed.lodz.pl (P.B.); 2Clinic of Thoracic Surgery, General and Oncological Surgery, USK im. WAM CSW, 90-549 Lodz, Poland; konradkedzia@gmail.com; 3Department of Clinical Pharmacology, Medical University of Lodz, 90-153 Lodz, Poland; lukasz.mokros@umed.lodz.pl

**Keywords:** OSAS, risk factors of OSAS, morning headaches, BMI

## Abstract

Morning headache is considered to be a symptom of obstructive sleep apnea syndrome (OSAS). Despite not being as common as excessive daytime sleepiness or unrefreshing sleep, it can similarly impair everyday activities. The aim of the present study was to evaluate the prevalence of and factors associated with morning headaches (MH) among patients referred for polysomnography due to suspected OSAS. This is a retrospective study on 1131 patients who underwent polysomnography between 2013 and 2015. Morning headaches (MH) were reported in 29% of them. In a logistic regression model, a rise in the n probability of MH was associated with female sex (odds ratio, OR, 1.38, 95% confidence interval, CI, 1.08–1.75), history of hypertension (OR 1.25, 95% CI 1.06–1.46), complaint on unrefreshing sleep (OR 1.42, 95% CI 1.19–1.70), choking at night (OR 1.25, 95% CI 1.05–1.49), and fall in total sleep time (OR 0.872 per each hour, 95% CI 0.76–0.99). The risk between MH and apnea–hypopnea index, blood oxygen saturation parameters or arousal index was found to be statistically insignificant. There is a lack of evidence that MH is associated with the severity of OSAS or nocturnal hypoxemia.

## 1. Introduction

In obstructive sleep apnea syndrome (OSAS), repeated episodes of obstructive apnea and hypopnea during sleep lead to sleep disruption by arousals. In consequence, patients suffer from excessive daytime sleepiness [1]. Other common OSAS symptoms include loud snoring, waking up with a choking or gasping sensation, sleepiness or lack of energy during the day, and morning headaches (MH). Recurrent arousals increase the risk of hypertension, cardiovascular, and cerebrovascular events [2,3]. The reported prevalence of OSAS ranges from 4% to 84% among men and from 2% to 61% among women [4,5].

MH has been considered to be a symptom of OSAS for more than a century [6,7]. The importance of the link is reflected in the 3rd edition of the International Classification of Headache Disorders since it provides diagnostic criteria for OSAS related headache. By definition, OSAS-related headache is present on awakening after sleep in a patient with diagnosed OSAS based on the apnea–hypopnea index of at least five episodes per hour of sleep (AHI > 5) [8]. It is also underlined that the cause–effect link between the MH and the presence of OSAS should be established [9].

MH can impair everyday activities similar to excessive daytime sleepiness and unrefreshing sleep, despite not being as common a symptom as the latter. Various pathophysiological theories have been proposed to explain the association between MH and OSAS. It is hypothesized that the sensation of pain appears due to consequences of repeating apneas, which include hypoxemia reflected by decrement in blood oxygen saturation (SpO_2_) and cerebral vasodilation due to a rise in carbon dioxide partial pressure, but also sleep fragmentation. However, the exact cause of this type of headache still remains unclear [6,7]. The most extensively studied epidemiological association in OSAS patients was between headaches and snoring [10,11]. MH is not specific and may complicate many other diseases such as anxiety and depression, circadian rhythm disorders, unspecified dyssomnia, and last but not least, arterial hypertension [11,12,13].

Therefore, the aim of this study was to assess the prevalence of MH and evaluate typical OSAS symptoms, risk factors, and respiration-related polysomnography (PSG) parameters as its predictors among patients referred to the sleep laboratory with a presumptive diagnosis of OSAS.

## 2. Materials and Methods

The study was conducted in the sleep center of the Department of Sleep Medicine and Metabolic Disorders, Medical University of Lodz, Poland. Clinical and polysomnographic data on 1181 patients were collected retrospectively from the beginning of 2013 to the end of 2015. In total, 172 patients were excluded based on the following exclusion criteria: less than 3 h of total sleep time (*n* = 41), central sleep apnea syndrome (*n* = 7), pure obesity hypoventilation syndrome (*n* = 4), poor signal quality of recorded channels (*n* = 13), lack of blood saturation parameters (i.e., basal SpO_2_, average SpO_2_ of desaturations (*n* = 107). After exclusion, 1009 patients (76.1% male) remained eligible for analysis (Figure 1). All patients were referred to the center due to the presumptive diagnosis of OSAS based on the aforementioned typical symptoms including unrefreshing sleep, snoring, and excessive daily sleepiness. Patient’s questionnaire included a question on the frequency of morning headaches. It was arbitrary assumed that if a patient reported at least one morning headache per week, they were assigned to the MH group. All patients underwent nocturnal diagnostic polysomnography (PSG), which is considered a diagnostic gold standard 12. PSG variables of interest included the apnea–hypopnea index (AHI), basal SpO_2_, mean SpO_2_ of desaturations, SpO_2_ nadir, arousal index (AI), total sleep time (TST), but also clinical ones of age, history of hypertension, body mass index (BMI), a score on the Epworth sleepiness scale, and complaints on unrefreshing sleep, snoring, choking at night, excessive daytime sleepiness, and observed apneas. Clinical variables from patients’ charts, and polysomnographic variables were used to create a database. A total of 1009 patients with all the necessary data including information regarding manifestation and frequency of morning headaches were eligible for further analysis.

All patients were admitted to the sleep center about 9 p.m. (±0.5 h) and underwent physical examination including measuring body weight and height to calculate BMI. Between 10 to 11 p.m., patients were ready to start polysomnography examination. Recorded channels comprised of electroencephalography (C4\A1, C3\A2), chin muscles and anterior tibialis electromyography, electrooculography, measurements of oro-nasal air flow (a thermistor gauge), snoring, body position (a gravitational gauge placed on the sternum), respiratory movements of the chest and abdomen (piezoelectric strain gauges), unipolar electrocardiogram, and hemoglobin oxygen saturation (SpO_2_) (Sleep Lab, Jaeger—Viasys, Hoechberg, Germany).

### 2.1. Compliance with Ethical Standards

The study was conducted as a project at the Medical University of Lodz. This study was conducted in accordance with the amended Declaration of Helsinki, and the Ethics Committee of the Medical University of Lodz approved the study protocol (RNN/23/15/KE; RNN/393/19/KE).

### 2.2. Data Availability Statement

The Excel data, used to support the findings of this study, may be released upon application to the Medical University of Lodz; the contact person is wojciech.kuczynski@umed.lodz.pl at the Department of Sleep Medicine and Metabolic Disorders

### 2.3. Statistics

The statistical analysis was conducted in STATISTICA 12 PL (Tulsa, OK, USA). Continuous variables of interest were characterized by mean values with standard deviations. The verification of normality of distribution of continuous variables was abandoned due to number of observations exceeding 50, based on the central limit theorem. Levene’s test was used to verify the hypothesis of homogeneity of variances between the compared groups. If the assumption was met, Fisher’s test was used in the analysis of variance, if otherwise, Welch’s test was used. A logistic regression model was constructed for the morning headache prediction analysis. Chi-square statistics were calculated to verify its goodness of fit and its results were presented as odds ratios (OR) with 95% confidence intervals (CI). Level of significance was established at α = 0.05

## 3. Results

Out of 1009 recruited individuals, 241 were women (23.9%). A total of 29.5% of the study group reported morning headaches (*n* = 298).

Patients with MH had lower AI (23.1 vs. 20.1, *p* = 0.01) and AHI (28.5 vs. 23.7, *p* = 0.01) than the no-headache (n-MH) group. There were no differences in age (52.0 vs. 52.1, *p* = 0.90), body mass index (BMI) (31.92 kg/m^2^ vs. 32.55 kg/m^2^, *p* = 0.15), total sleep time (TST) (5.9 h vs. 5.7 h, *p* = 0.06), basal SpO_2_ (92.2% vs. 92.0, *p* = 0.48), mean SpO_2_ of desaturations (86.9 vs. 87.1, *p* = 0.61), and SpO_2_ nadir (77.8% vs. 78.2%, *p* = 0.60). All clinical variables are listed in the Table 1.

Women, history of hypertension, complaints on unrefreshing sleep, and choking at night were more prevalent in the MH group vs. the n-MH group (Table 2).

We verified whether and how the above variables modified the probability of MH as a complaint. The constructed logistic regression model was fit to the data (Chi^2^ = 18.85, *p* < 0.001). Statistically significant risk factors for MH included: female sex (OR 1.375, 95% CI 1.08–1.75, *p* < 0.01), history of hypertension (OR 1.25, 95% CI 1.06–1.46; *p* = 0.006), complaints on unrefreshing sleep (OR 1.424; 95% CI 1.19–1.70; *p* < 0.001), choking at night (OR 1.25; CI 95% 1.05–1.49; *p* = 0.012) and TST (OR 0.87; CI 95% 0.76–0.99; *p* = 0.047). Interestingly, a rise in the probability of reporting MH was independent from basal SpO_2_ (OR 0.95; 95% CI 0.89–1.01; *p* = 0.066), mean SpO_2_ desaturations (OR 1.01; 95% Cl 0.99–1.04; *p* = 0.37), SpO_2_ nadir (OR 0.99, 95% CI 0.97–1.01), but also AHI (OR 0.99; 95% CI 0.98–1.00; *p* = 0.055) or AI (OR 0.99; 95% CI 0.89–1.01; *p* = 0.59).

The association between the occurrence of MH and the other variables in the model (i.e., age, BMI, Epworth sleepiness scale score, smoking, snoring, observed apnea and sleepiness during the day) was not statistically significant (Table 3).

## 4. Discussion

Previous studies have indicated that morning headaches may concern 7.6% of the general population [14]. Our results show that MH may be frequently reported among patients referred for diagnostics with a presumptive diagnosis of OSAS, but to our surprise, it may not necessarily be linked to OSAS itself. Neither AHI, nor the AI or SpO_2_ parameters were associated with the probability of MH in a statistically significant manner. This result partially confirms previous findings from case–control studies, which concluded the lack of relationship between the severity of OSAS and headaches [15,16]. On the other hand, Russell et al. found associations between MH and time below 90% SpO_2_, level of average oxygen desaturation, and of the lowest SpO_2_, but not average SpO_2_. Kristiansen et al. suggested that nocturnal hypoxia may play an important role in the pathophysiology of morning headaches, but only among susceptible persons [16,17]. This problem requires further investigation. In the studied group, a clinical profile of a patient admitted to the sleep center and reporting morning headaches partially overlapped the features linked to OSAS (i.e., it comprised obesity and history of hypertension). On the other hand, there were more women in the MH group than in the n-MH group. This result is concordant with data that morning headaches are generally more prevalent among women than men [14]. However, it should be remembered that men had a higher rate of OSAS compared to women. This ratio is approximately 2:1 or 3:1, but it diminishes when women at a post-menopausal age are considered [18,19].

Despite the high prevalence of morning headaches in the investigated population, they may not be associated with OSAS. Thus, the diagnosis of “sleep apnea headache” was introduced to differentiate it from a general category of morning headache. More detailed information on the headache including its duration, frequency, character, and intensity might be required during its assessment as an OSAS symptom [15,17]. Chronic headaches can be regarded as soft signs of a sleep disorder, not necessarily OSAS [20,21]. The results of this study are in line with this statement, since complaints on unrefreshing sleep, choking at night, and decreased sleep duration were found to be significantly associated with MH incidence. Those three factors are not specific and may additionally be related to insomnia, anxiety, or depressive disorders [22]. Those diagnoses are more prevalent among women than men and female sex was also found to be a risk factor for MH in this study [23]. Furthermore, circadian rhythm disruption is recognized to affect systemic blood pressure and arterial hypertension was also associated with MH [24]. The complex relationship between sleep disturbances and headaches still requires further investigation. However, for clinical practice, it is crucial to perform a complex differential diagnosis, possibly including polysomnography, when a patient presents both of these complaints [20,22].

It should be noted that there are not many publications investigating morning headache and its determinants in a high-risk OSAS population. Another key strength of the current study is the size of the study group, since few similar studies have included more than 1000 patients who underwent nocturnal polysomnography. On the other hand, each patient had only a single-night PSG. It may be considered a limitation, since an internight variability of the results has been previously observed, particularly in mild OSAS cases [25]. However, it should be noted that PSG is a complex and timely procedure. Thus, in most cases, clinical decisions concerning diagnosis and therapy are based on results from one night. One of the weaknesses of the current study is its selection bias, as our results apply to preselected subjects (i.e., the referred group of Caucasian patients with high pre-test probability of disease), not a representative sample of the general adult population.

## 5. Conclusions

Based on hereby study, there is a lack of evidence that MH is associated with the severity of OSAS, or nocturnal hypoxemia. Those results confirm the need of “sleep-related headache” as a separate diagnostic class, since a general complaint on morning headache is highly unspecific, yet it may indicate other sleep disturbances.

## Figures and Tables

**Figure 1 brainsci-10-00057-f001:**
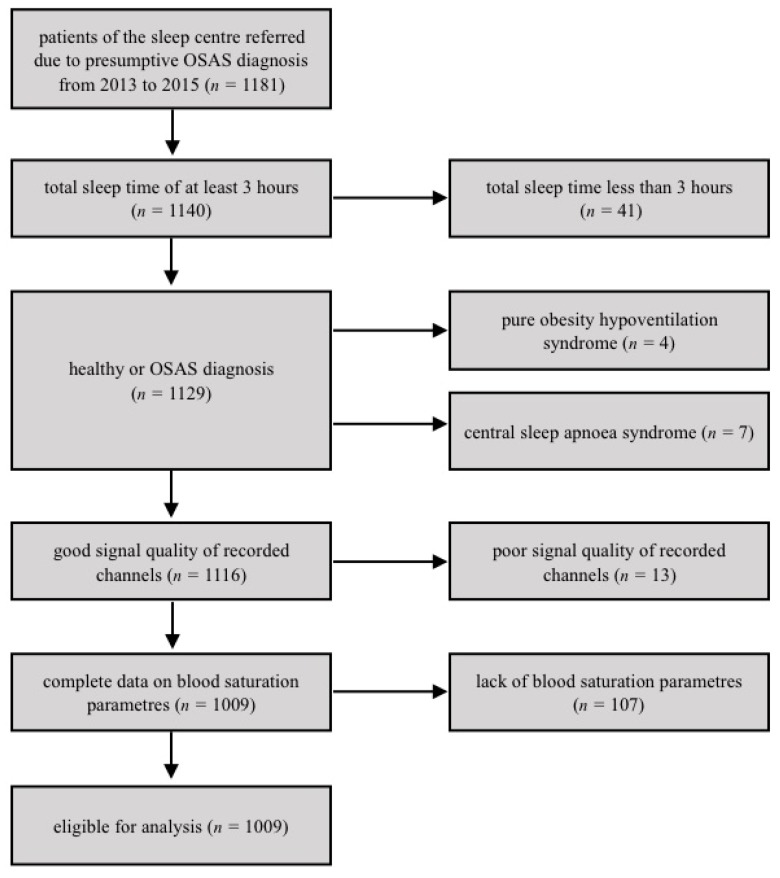
Flow chart showing selection of the patients eligible for the analysis in the studied group of patients with suspected obstructive sleep apnea syndrome (OSAS). *n*—number of observations.

**Table 1 brainsci-10-00057-t001:** Summary of the clinical and sleep study variables.

MH ≥ 1 per Week						
	No MH (*n* = 711)		MH (*n* = 298)			
	M	SD	M	SD	F	*p*
age	52.00	12.09	52.11	11.31	0.02	0.90
BMI	31.92	5.83	32.55	6.58	2.06	0.15
Neck circumference	41.92	4.12	41.45	4.54	2.61	0.11
TST	5.85	1.09	5.71	1.09	3.47	0.06
Arousal index	23.11	17.88	20.13	16.42	6.57 *	0.01
AHI	28.46	26.94	23.66	25.78	7.11 *	0.01
Basal SpO_2_	92.21	3.52	92.04	3.60	0.50	0.48
Mean SpO_2_ of desaturations	86.87	7.53	87.13	7.35	0.25	0.61
SpO_2_ nadir	77.79	10.51	78.17	11.37	0.27	0.60
% of time below 90%	19.00	23.00	17.00	22.00	2.31	0.13
Epworth sleepiness scale	8.39	4.47	8.49	4.22	0.11	0.74

F—statistics in the analysis of variance or Welch’s test depending on Levene’s test result, *p*—probability in the statistical test, *—the criterion of homogeneity of variance was not fulfilled, Welch’s test was used in those cases, MH—morning headache, M—mean value, SD—standard deviation, BMI—body mass index, TST—total sleep time, AHI—apnea–hypopnea index, SpO_2_—blood oxygen saturation.

**Table 2 brainsci-10-00057-t002:** Prevalence of women, smoking, hypertension, and chosen clinical symptoms related to sleep-disordered breathing in the studied sample.

	No MH (*n* = 711)		MH (*n* = 298)		Chi^2^	*p*
	*n*	%	*n*	%		
Female	143	20.11%	98	32.89%	18.846	< 0.001
Hypertension	388	54.57%	194	65.10%	9.538	0.002
Smoking	349	49.09%	133	44.63%	1.670	0.196
Snoring	565	79.47%	226	75.84%	1.631	0.202
Observed apneas	524	73.70%	209	70.13%	1.343	0.247
Excessive sleepiness	240	33.76%	99	33.33%	0.017	0.897
Unrefreshing sleep	108	15.19%	75	25.17%	14.081	< 0.001
Choking at night	120	16.88%	75	25.17%	9.256	0.002

*n*—number of observations, %—percentage of observations, Chi^2^—Chi^2^ statistical test result, *p*—probability in the Chi-square test for the contingencies.

**Table 3 brainsci-10-00057-t003:** Logistic regression results for prediction of morning headaches in the studied sample.

	OR	OR 95% CI		Wald	*p*
Female sex	1.375	1.080	1.751	6.694	0.010
Hypertension	1.246	1.064	1.461	7.417	0.006
Smoking	0.951	0.822	1.100	0.453	0.501
Snoring	0.866	0.729	1.028	2.694	0.101
Observed apneas	1.077	0.912	1.271	0.758	0.384
Sleepiness	0.937	0.800	1.097	0.654	0.419
Unrefreshing sleep	1.424	1.192	1.700	15.231	0.000
Choking at night	1.253	1.051	1.495	6.313	0.012
Age	0.987	0.974	1.001	3.322	0.068
BMI	1.017	0.978	1.057	0.689	0.406
Neck circumference	1.002	0.939	1.070	0.005	0.943
TST	0.872	0.763	0.998	3.952	0.047
Arousal index	0.997	0.984	1.009	0.292	0.589
AHI	0.990	0.979	1.000	3.675	0.055
Basal SpO_2_	0.947	0.894	1.004	3.373	0.066
Mean SpO_2_ of desaturations	1.013	0.985	1.041	0.809	0.368
SpO_2_ nadir	0.988	0.966	1.010	1.148	0.284
% time below 90%	0.619	0.209	1.832	0.751	0.386
Epworth sleepiness scale score	1.012	0.978	1.047	0.484	0.487

OR—odds ratio, 95% CI—95% confidence interval for odds ratio; Wald—Wald statistics for a single parameter in the model, *p*—probability in the Wald’s statistical test, BMI—body mass index, TST—total sleep time, AHI—apnea–hypopnea index, SpO_2_—blood oxygen saturation.

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
