# Peer review of "Morning Headache as an Obstructive Sleep Apnea-Related Symptom among Sleep Clinic Patients—A Cross-Section Analysis"

_brainsci, 2020, doi:10.3390/brainsci10010057_

Round 1

Reviewer 1 Report

Overall: The discussion section could be amplified

Detailed comments:

1) Even this study is based on a large population, a possible limitation is the evaluation of one-night PSG (high inter-night variability)

2) By PSG the authors evaluated also the leg movements, but there are no data on PLM. This could be important in relation to the finding of "unrefreshing sleep" that resulted significantly related to the headache. The authors could speculate about the relationship between PLM related and unrelated to sleep apnea.

3) Complaint on unrefreshing sleep and fall in total sleep time could be related to restless legs syndrome (RLS) that is well known to be in comorbidity with OSA: did the authors consider this possible comorbidity ? Interestingly,  RLS is more prevalent in women....

4) The time of SaO2 below 90% is surprisingly low in the studied population by considering the mean BMI (>31 in both subgroups). Please, comment this finding.

Author Response

We would like to thank the reviewers for their valuable comments and the opportunity to improve our manuscript. The language of the manuscript was reviewed and improved.

Response to Reviewer 1

Overall: The discussion section could be amplified

Answer to overall comment: We added additional paragraphs and correction in accordance with Reviewers’ comments. It should be noted that there is scarce data published considering the investigation of the association between morning headaches and OSAS.

Detailed comments:

Even this study is based on a large population, a possible limitation is the evaluation of one-night PSG (high inter-night variability)

Answer to comment 1: We agree that inter-night variability may be a confounding factor. A proper update of the paragraph concerning limitations has been introduced.

By PSG the authors evaluated also the leg movements, but there are no data on PLM. This could be important in relation to the finding of "unrefreshing sleep" that resulted significantly related to the headache. The authors could speculate about the relationship between PLM related and unrelated to sleep apnea. Complaint on unrefreshing sleep and fall in total sleep time could be related to restless legs syndrome (RLS) that is well known to be in comorbidity with OSA: did the authors consider this possible comorbidity? Interestingly,  RLS is more prevalent in women....

Answer to comment 2 and 3: In Methods section, the whole standard procedure of PSG was described, which is why we included information that leg movement was recorded. However, it was not considered variable of interest, similarly to REM and non-REM phase of the sleep (also recorded during PSG procedure). We agree that PLM (or RLS) may be related to unrefreshing sleep, but so can the coexistence of depressive or anxiety symptoms or disrupted sleep architecture. Including more well-known correlates of unrefreshing sleep would extensively increase the number of variables, which could make the analysis incomprehensible. Thus, we decided to focus only on certain aspects of the clinical evaluation. The objective of the study was to concentrate on typical OSAS symptoms, risk factors and respiration-related PSG parametres (i.e. AHI, saturation, desaturation) and assess whether they are related to the incidence of MH in this highly specific population. The aim of the study (last paragraph of introduction) was modified to make it more precise and adequate to the topic. Further studies should consider more possible confounding factors, but this should be followed by an increase in the number of observations.

4) The time of SaO2 below 90% is surprisingly low in the studied population by considering the mean BMI (>31 in both subgroups). Please, comment on this finding.

There has been a mistake in data presentation: the description of the row mentioned “% of time below 90%”, while the values were presented as decimals. Table 1 was properly improved, which should resolve any concerns about this parameter. Also, the rest of the data was verified whether their presentation is correct.

Reviewer 2 Report

This is very nice review of a topic that would be of great interest to the medical and sleep community. 

Comments

1. Please add the aim of this study at the end of introduction section

2. Please add the flow chart in method about the procedures 

3. Please add in results the parameters of pulmonary fuction (FEV1, PEF, etc)

4. Please add abbreviations at the end of each table

5. In results, tables and text have same informations. Pleace replace

6. Please add a limitations and cocnlusions section 

Author Response

We would like to thank the reviewers for their valuable comments and the opportunity to improve our manuscript. The language of the manuscript was reviewed and improved.

Response to Reviewer 2

Please add the aim of this study at the end of the introduction section

Answer to comment 1: The aim of the study was already included, yet we made it more precise in response to the comments.

Please add the flow chart in method about the procedures 

Answer to comment 2: The flow chart has been added accordingly.

Please add in results the parameters of pulmonary function (FEV1, PEF, etc)

Answer to comment 3: The pulmonary function (meant by parametres like FEV1 or PEF) was not assessed in the study group, since such assessment is not a part of standard PSG procedure. This is a retrospective analysis and the aforementioned parameters were not considered variables of interest. We would like to underline that the focus of this study was on certain PSG findings, typical OSAS symptoms and risk factors.

Please add abbreviations at the end of each table

Answer to comment 4: The abbreviations were added accordingly.

In results, tables and text have same informations. Pleace replace

Answer to comment 5: Although it may be considered redundant, we decided to leave the presentation of results in the current form, thus descriptive statistics and regression analysis parametres remain in both text and tables.

Please add a limitations and cocnlusions section 

Answer to comment 6: The limitations description was extended and contained in the last paragraph of the discussion. The Conclusions section has been added accordingly.

Round 2

Reviewer 2 Report

The manuscript is acceptable.